# Evaluation of an Inpatient Psychiatric Mother-Baby Unit Using a Patient Reported Experience and Outcome Measure

**DOI:** 10.3390/ijerph19095574

**Published:** 2022-05-04

**Authors:** Grace Branjerdporn, Carly Hudson, Roy Sheshinski, Linda Parlato, Lyndall Healey, Aleshia Ellis, Alice Reid, Catherine Finnerty, Rachelle Arnott, Rebecca Curtain, Miranda McLean, Snehal Parmar, Susan Roberts

**Affiliations:** 1Mental Health and Specialist Services, Gold Coast Hospital and Health Service, Gold Coast, QLD 4215, Australia; carly.hudson@health.qld.gov.au (C.H.); sheshinski@protonmail.com (R.S.); linda.parlato@health.qld.gov.au (L.P.); lyndall.healey@health.qld.gov.au (L.H.); aleshia.ellis@health.qld.gov.au (A.E.); shannon.reid2@health.qld.gov.au (A.R.); catherine.finnerty@health.qld.gov.au (C.F.); rachelle.arnott@health.qld.gov.au (R.A.); rebecca.curtain@health.qld.gov.au (R.C.); miranda.mclean@health.qld.gov.au (M.M.); snehal.parmar@health.qld.gov.au (S.P.); susan.roberts2@health.qld.gov.au (S.R.); 2Mater Young Adult Health Service, Mater Hospital, South Brisbane, QLD 4101, Australia

**Keywords:** mother-baby unit, perinatal mental health, inpatient, patient-reported outcomes, patient-reported experience, service evaluation

## Abstract

Understanding the patient experience of admission to a psychiatric mother-baby unit (MBU) informs service improvement and strengthens patient-centered care. This study aims to examine patients’ experience, satisfaction, and change in mental health status related to MBU admission. At discharge, 70 women admitted to a public MBU completed the Patient Outcome and Experience Measure (POEM), rated the usefulness of therapeutic groups, and provided written qualitative feedback. Paired sample t-tests, correlations, and thematic content analysis were completed. Women were highly satisfied with the level of care and support received, particularly for those who were voluntarily admitted. Women reported an improvement in mental health from admission to discharge. Women appreciated the staff’s interpersonal skills, provision of practical skills, education, advice, support from other women, and therapeutic groups offered. Women suggested improvements such as having greater food choices, more MBU beds, more group sessions, family visitations, which had been restricted due to COVID-19, environmental modifications, and clarity of communication surrounding discharge. This study highlights the benefits of MBUs and the specific aspects of care that are favorable in treating women with mental illnesses who are co-admitted with their baby in an MBU.

## 1. Introduction

Estimates suggest that at least 20% of women have significant mental health problems during the first twelve months after birth [1]. The postnatal period is a particularly vulnerable period for women to develop or experience recurrence of psychiatric illnesses [2], with suicide being the leading cause of maternal deaths in Queensland, Australia [3], and the second leading cause of maternal deaths in the United States, between one month and one year postpartum [4]. Left untreated, maternal mental illnesses have been linked with a range of longer-term adverse outcomes such as difficulties with daily functioning, decreased mother–infant bonding, and suboptimal infant development [5].

For women with severe postnatal mental illness requiring inpatient mental health treatment, co-admission to a specialist mother and baby unit (MBU) is considered best practice to treat and manage maternal mental illness whilst avoiding separation from her infant [6]; therefore, providing access to quality care and specialized perinatal and infant mental health treatment to women who have severe or complex mental illness in the postpartum year is vital, as it ensures that the needs and additional risks to women and infants will be met. The MBU examined in the present study comprises a multidisciplinary treating team, consisting of psychiatric medical officers, mental health nursing staff, child health nursing staff, pediatric medical officers, and a variety of allied health staff (e.g., occupational therapist, physiotherapist, dietitian, social worker, psychologist, pharmacist, and an allied health professional specializing in infant mental health). A range of severe mental illnesses are treated within an MBU, such as severe depressive disorders, eating disorders, bipolar affective disorders, and borderline personality disorders. A holistic approach to recovery is adopted, which aims to improve maternal mental health, as well as foster a positive mother and baby bond, secure infant attachment, and parenting skills [7]. 

Although MBU admission has been found to be effective in improving maternal mental health and maternal–infant attachment [8], further research is required which uses patient-reported experience measures (PREMs), patient-reported outcome measures (PROMs), and qualitative feedback to evaluate MBU admissions. PREMs measure a patient’s perception of their experience within a health care system or service, focusing on specific elements of care related to relational aspects, identifying the patients' experience of their relationships during treatment (e.g., staff communication), and functional aspects that focus on more practical issues (e.g., adequacy of facilities) [9]. PROMs measure clinical outcomes of a healthcare service and are completed by patients to ascertain perceptions of their change in health status due to the treatment [9]. Finally, qualitative approaches enable patients to share their perspectives and gain rich information about their experiences [10].

Only one internally published report called the Patient Outcome and Experience Measure (POEM) [11] has used a PREM and a PROM to evaluate the functional and relational aspects of an inpatient psychiatric MBU admission. When exploring the literature more broadly, patient satisfaction of MBU admissions has been examined using a non-standardized telephone-based survey [12], and the Mother and Baby Unit Satisfaction Questionnaire [13,14], which identified aspects that women were satisfied and dissatisfied with in relation to therapeutic activities, involvement in care, family inclusion, environmental considerations, and communication with staff [12,13,14]. Understanding and integrating the patient’s perspective into service delivery is important in co-designing services, and it underscores the value of partnering with those who have a lived experience of mental illness [15].

This study aims to explore the patient experience of an MBU admission. There are four key research questions: firstly, to understand the experience of admission to an MBU, including the usefulness of therapeutic groups; secondly, to examine the change in perceived mental health from admission to discharge; thirdly, to evaluate demographic variables associated with patient experience; fourthly, to examine qualitative feedback given by patients prior to discharge. It is hypothesized that the MBU experience will be viewed positively in most aspects, women will perceive that their mental health has improved due to the admission, that some (demographic or diagnostic) characteristics of patients may be related to their experience, and that qualitative feedback will highlight both areas of strength and improvement for the MBU service.

## 2. Materials and Methods

### 2.1. Study Setting

To assess and treat women with severe mental illness in the first year postpartum, an MBU inpatient facility was established in Queensland, Australia. This statewide public service admits four mothers and their babies who are under one year old. The Lavender MBU is an acute mental health inpatient unit that provides a holistic model of care to women admitted, with the aim of improving maternal mental health whilst fostering positive mother–infant attachment and parenting confidence, supported by a large multidisciplinary team of medical, nursing, and allied health professionals. Following discharge, women are linked with a range of community mental health and psychiatric services [16]. Women eligible for admission require inpatient treatment for a mental health condition that cannot be managed in the community, have an infant under 12 months old, reside in Queensland (Australia), and are not homeless or at risk of homelessness. Women not eligible for admission are those that require detox for a substance or alcohol use disorder, or have a baby with an infectious disease.

**Table 1 ijerph-19-05574-t001:** Demographic Characteristics of the Present Study Sample.

Variables	*n*	%	M	SD	Min	Max
Mother’s age (years)	69	98.57	29.72	5.45	19	42
Baby’s age (weeks)	69	98.57	17.10	13.12	1	52
Length of stay (days)	69	98.57	22.25	11.64	2	61
Socioeconomic status ^1^	69	98.57	65.25	24.19	2	98
**Aboriginal and Torres Strait Islander Status**	69	98.57				
Neither Aboriginal nor Torres Strait Islander	64	92.75				
Aboriginal and/or Torres Strait Islander	5	7.25				
**Marital status**	69	98.57				
Married	43	62.32				
Never Married	25	36.20				
Separated	1	1.45				
**Involuntary Admission**	69	98.57				
Voluntary	57	82.61				
Involuntary	12	17.39				
**Baby’s sex**	69	98.57				
Male	35	50.72				
Female	34	49.28				
**Country of birth**	69	98.57				
Australia	57	82.61				
Asia	5	7.25				
United Kingdom	4	5.79				
New Zealand	3	4.35				
**Primary psychiatric diagnosis**	69	98.57				
Depressive disorder	34	49.27				
Anxiety disorder	9	13.04				
Personality disorder	8	11.59				
Bipolar affective disorder	7	10.14				
Psychotic disorder	6	8.69				
Anorexia nervosa	5	7.25				

^1^ Zip code was used to rank the mother’s social-economic status percentile in the state of Queensland, according to the Index of Relative Socioeconomic Advantage and Disadvantage (IRSAD) [17]. Note. Although the POEM was completed by 70 women, only demographic details of 69 participants were available, given that one survey was anonymously completed.

### 2.2. Participants

All women who were accepted for admission based on the criteria above were eligible to participate. Between January 2019 and January 2021 (24 months), 70 women completed the POEM, which was 55% of the women (*n* = 126) admitted during the study period. Women were aged between 19 and 42 years, with an average age of 29.72 years (SD = 5.45). Babies were, on average, 17.10 weeks old (SD = 13.12), and 82.61% women (*n* = 57) were born in Australia, with 7.25% (*n* = 5) from Asia, 5.79% (*n* = 4) from the United Kingdom, and 4.35% from New Zealand (*n* = 3). Moreover, 7.25% (*n* = 5) of the women identified as either Aboriginal or as Torres Strait Islander. All women had a primary psychiatric diagnosis which included the following: depressive disorder (49.27%, *n* = 34), anxiety disorder (13.04%, *n* = 9), personality disorder (11.59%, *n* = 8), bipolar affective disorder (10.14%, *n* = 7), psychotic disorder (8.69%, *n* = 6), and anorexia nervosa (7.25%, *n* = 5). The average length of stay was 22.25 days (SD = 11.64), with a range of 2 to 61 days. The socioeconomic status of the women was, on average, in the 65.25th percentile of Australia (SD = 24.19). Demographics of the participants are reported in Table 1.

### 2.3. Procedure

This study utilized a cross-sectional mixed-methods survey completed at discharge, which is a commonly used and well-recognized method to assess patient satisfaction with admission [13,14]. Within 48 h prior to discharge, a nursing staff member distributed the self-reported questionnaire and explained that the questionnaire guided service development. It was also outlined that participation was voluntary and did not affect the clinical care provided. Mental health nursing staff were available as required to provide emotional and practical support to women as they were completing the questionnaire. The survey contained the POEM and multiple-choice questions about the usefulness of therapeutic groups. There were also free-text questions about areas of strengths and improvements, which could be anonymously completed if desired. The data was later scanned into hospital medical records, extracted by the research team, and deidentified for analysis. The project was exempt from ethical review by the Gold Coast Health Human Research Ethics Committee (LNR/2018/QGC/47991), as this was considered a quality assurance project with data collection being part of routine care. 

### 2.4. Measures

#### 2.4.1. Demographics

Demographic information, including the participants’ zip code, maternal and infant age, marital status, country of birth, primary psychiatric diagnosis (based on the category of disorder), length of stay, involuntary admission status, and First Nations Status (i.e., Aboriginal and/or Torres Strait Islander), were extracted from the hospital medical records. To represent the participant’s social-economic status, the Index of Relative Socioeconomic Advantage and Disadvantage (IRSAD) percentile rank within Australia was used [17]. The IRSAD corresponds to a participant’s zip code and indicates the relative economic and social advantage and disadvantage of people living within a particular area [17].

#### 2.4.2. Patient Outcome and Experience Measure (POEM)–Inpatient Measure

The POEM–Inpatient Measure is specific for psychiatric, inpatient MBUs and is composed of 20 items with a PROM and a PREM section [18]. For the PROM, two questions examined the self-perceived mental health status at admission and discharge (i.e., “When I first came into contact with the service, I was…”, “When I was discharged from the service, I was…”), using a 5-point Likert scale (1 = extremely unwell, 2 = very unwell, 3 = unwell, 4 = well, 5 = very well). For the PREM, there were 18 questions measuring the patient’s view of their experience in the service using a 4-point Likert scale (1 = strongly agree, 2 = agree, 3 = disagree, 4 = strongly disagree). Twelve questions focused on relational aspects of inpatient service (e.g., “Staff gave me the right amount of support and care”, “Staff were not very sensitive to my needs”), with five questions negatively worded and reverse coded. Six questions examined functional aspects of the inpatient service (e.g., “The unit provided a good place for my baby to be with me”, “The food provided was not acceptable to me”). The POEM has been recommended by the Royal College of Psychiatry’s Centre for Quality Improvement (CCQI) to support service improvement over time [18] and has been used in the United Kingdom [11]. Internal consistency was adequate for the overall score (α = 0.87), relational aspects of inpatient service (α = 0.89), and functional aspects of the inpatient service (α =0.74). 

#### 2.4.3. Additional Therapeutic Groups Usefulness Questions

An additional six questions were included that asked the patient to rate their perceived usefulness of therapeutic groups facilitated within the MBU. This was measured using a 3-point Likert scale (1 = very useful, 2 = somewhat useful, 3 = not useful at all, or did not participate).

#### 2.4.4. Qualitative Free-Text Responses

The questionnaire also had a provision for the patients to write additional free-text comments regarding how their stay in the MBU could be improved, and any positive feedback about their experience.

### 2.5. Data Analysis

#### 2.5.1. Quantitative Statistical Analysis

Data analysis was conducted using the IBM Statistical Package for Social Science Version 27. Descriptive statistics, including minimum, maximum, mean, and standard deviations for continuous variables and frequencies and proportions for categorical variables were computed for demographic variables (Table 1). The difference in the patient’s self-rated mental health status at admission and discharge was assessed using a paired samples *t*-test (Table 2). To understand the clinical and demographic variables correlated with this change, Pearson correlations were completed between the change in mental health status from admission to discharge, and the following variables: maternal age, infant age, socioeconomic status percentile, marital status, first nation status, infant sex, length of stay, primary diagnosis when dummy coded, mental health act status. Those who agreed (i.e., strongly agree and agree) and disagreed (Strongly Disagree and Disagree) with POEM statements were aggregated, and proportions of items and averages of subscales were generated (see Table 3). Those who found the therapeutic group programs to be “very useful” or “somewhat useful” were aggregated as “useful”, and proportions were calculated. Additionally, proportions were computed for those who rated the programs as “did not participate” or “not useful” (see Table 4). As the PREM component was not normally distributed, Spearman’s correlations were conducted between demographics with POEM subscales (see Table 5). Marital status was aggregated into a dichotomous variable (e.g., not married and separated = 0; married = 1). An alpha level of 0.05 was used to assess statistical significance in correlations. 

#### 2.5.2. Qualitative Analysis

Thematic content analysis was used to examine the qualitative feedback provided by patients at the conclusion of the questionnaire (see Table 6). The Braun and Clarke method was used to guide the thematic content analysis [19]. The responses were independently coded by two researchers who then met to discuss, compare, and refine themes until full consensus was attained. Cross-tabulations with frequencies and proportions were completed to examine the nature of feedback (positive vs. suggestion for improvement) based on the Mental Health Act status (voluntary vs. involuntary).

**Table 2 ijerph-19-05574-t002:** Paired Samples *t*-Test Between Maternal-Rated Mental Health Status at Admission and Discharge with 1 = Extremely Unwell and 5 = Very Well.

	Admission		Discharge			
	*M*	*SD*	*M*	*SD*	*df*	*t*
Maternal-rated mental health status	2.05	0.84	4.16	0.72	68	−19.23 **

*****p*< 0.01; *M* = Mean; *SD* = standard deviation; *df* = degrees of freedom; *t* = *t-* statistic.

## 3. Results

### 3.1. PROM Rating of Mental Health Status at Admission to Discharge

A paired-samples *t*-test was conducted to compare patients’ perception of mental health status from admission to discharge. There was a significant increase in the mental health scores between admission (*M* = 2.1, *SD* = 0.84) and discharge (*M* = 4.16, *SD* = 0.72), *t* (68) = −19.23, *p* < 0.01, as seen in Table 2.

When correlations were completed between change in mental health status from admission to discharge, using a range of variables (maternal age, infant age, socioeconomic status percentile, marital status, first nation status, baby sex, length of stay, primary diagnosis, and mental health act status), none of the correlations were statistically significant. This indicates that patients perceive an improvement in mental health status from admission to discharge, regardless of any clinical and demographic characteristics.

**Table 3 ijerph-19-05574-t003:** Percentage of Agreement (Strongly Agree and Agree) and Disagreement (Strongly Disagree and Disagree) of Maternal-Rated Items from the POEM.

Items		Disagreement (%)	Agreement (%)
**Relational aspects of inpatient service**		*M* = 4.65	*M* = 95.35
	3. Staff communicated with others involved in my care. ^1^	7.14	92.86
	4. Staff gave me the right amount of support and care.	2.86	97.14
	5. I got help quickly enough after referral. ^1^	5.71	94.29
	6. Staff listened to me and understood my problems.	2.86	97.14
	7. Staff involved me enough in my care and treatment. ^1^	4.29	95.71
	8. The service provided me with the information I needed.	1.43	98.57
	9. Staff were very sensitive to my needs. ^1^	4.29	95.71
	10. Staff helped me understand my illness/difficulties.	5.71	94.29
	11. Staff were very sensitive to the needs of my baby. ^1^	7.20	92.80
	12. Staff helped me be more confident with caring for my baby.	5.71	94.29
	13. The service involved other relevant people in a helpful way.	5.71	94.29
	14. I would recommend this service to others.	2.86	97.14
**Functional aspects of the inpatient service**		*M* = 6.99	*M* = 91.90
	15. The unit was clean and hygienic.	1.43	98.57
	16. The unit provided a good place for me to recover. ^1^	2.90	97.10
	17. The unit provided helpful activities and therapies. ^1^	4.29	95.71
	18. The unit provided a good place for my baby to be with me.	2.90	97.10
	19. The unit supported me in my contact with family and friends.	2.86	97.14
	20. The food provided was acceptable to me. ^1^	27.54	72.46

^1^ Statement is worded positively in table but negatively in the original patient-reported questionnaire; items start at question 3, and questions 1 and 2 relate to the PROM questions.

### 3.2. PREM Ratings

As displayed in Table 3, relational aspects of the inpatient service were rated between 92.86% and 98.57% for level of agreement, with 95.35% agreement on average, indicating overall favorable perceptions. Moreover, 97.14% reported that they would recommend this service to others. Most of the functional aspects of the inpatient service were rated between 95.71% and 98.57% of agreement, suggesting high satisfaction; however, acceptability of the food (item 20) was rated with 72.46% agreement. Overall, the functional aspects of the inpatient service were rated positively, with 91.90% agreement.

**Table 4 ijerph-19-05574-t004:** Maternal-Rated Usefulness of Specific Therapeutic Group Program in The Mother-Baby Unit.

Therapeutic Programs	Description of Group	Lead Allied Health Discipline at Lavender	Not Participated (%)	Not Useful (%)	Useful ^^^ (%)
Sensory modulation	Sensory modulation education is provided to the group members to help them work on identifying their triggers and early warning signs of dysregulation. Mothers’ sensory preferences are assessed and are provided with education on using sensory modulation strategies to support optimal arousal. Sensory tools are trialled and selected for use throughout admission and post-discharge.	Occupational Therapist	7.58	0.00	92.42
Baby play	Mothers play with their baby using age-appropriate activities and games such as nursery rhymes, bubble blowing, and “tummy” time. Baby massage is taught to promote infant development and maternal–infant attachment. Staff provide education, encouragement, modelling, and practice.	Occupational Therapist, Physiotherapist, Social Worker, Infant Mental Health Therapist	12.12	1.52	86.36
Pharmacotherapy group (‘Medwise Group’)	Mothers ask questions and learn about the role of medications, different types of modifications and their side effects, and other issues around medication management. For example, topics discussed include breastfeeding while on an antidepressant.	Pharmacist	7.35	7.35	84.85
Mindfulness practice	Mothers practice mindfulness exercises which help to ‘ground’ the mothers and may be applied when with the baby (e.g., mindfulness during baby bathing, five senses technique).	Psychologist, Social Worker	10.61	6.06	83.33
Mother and baby relationship	Mothers are provided education about concepts related to Circle of Security, attachment, and self-care.	Social Worker, Infant Mental Health Therapist	19.68	0.00	80.32
Healthy lifestyle	Mothers complete meal and snack preparation for themselves and their baby, and are provided education about healthy eating, meal planning for the family, budgeting, baby nutrition and food, and mood relationship. They practice mindful eating skills.	Dietitian, Occupational Therapist, Physiotherapist	16.70	3.00	80.30
Mother and baby exercise	Mothers engage in exercises based on stretching, strengthening, and cardiovascular fitness, while safely involving their baby. These exercises aim to provide mothers with skills to exercise whilst interacting with their baby.	Physiotherapist	16.70	3.00	80.30
Positive coping strategies	Mothers are educated on and provided with the opportunity to practice positive coping strategies using compassion-centered therapy, dialectical behavior therapy, acceptance and commitment therapy, and cognitive behavioral therapy.	Psychologist	22.73	0.00	77.27

### 3.3. Usefulness of Therapeutic Programs

As displayed in Table 4, the therapeutic group programs led by a range of allied health were rated positively by those who participated (Range: 77.27–92.42%, Mean = 83.14%). Of the eight therapeutic programs, sensory modulation (92.46%), baby play (86.36%), and pharmacotherapy group (84.85%) had the highest ratings for level of usefulness. 

**Table 5 ijerph-19-05574-t005:** Spearman Correlations Between Demographic and Diagnostic Variables with PREM Subscales.

Demographic Variables	Relational Aspects	Functional Aspects
Mother’s age	0.08	−0.03
Length of stay (days)	0.04	0.01
Marital Status ^1^	0.01	−0.06
Socioeconomic status (state percentile) ^2^	0.08	0.20
Voluntarily admitted ^3^	0.33 ^**^	0.25 ^*^

^1^ Not Married and Separated = 0, Married = 1; ^2^ Zip code was used to rank the mother’s social-economic status percentile in the state of Queensland, according to the Index of Relative Socio-economic Advantage and Disadvantage (IRSAD); ^3^ Involuntary admission = 0, Voluntary admission = 1. * *p* < 0.05; ** *p* < 0.01.

### 3.4. Correlation between PREM Subscales and Demographic Variables

As shown in Table 5, voluntary admission was significantly positively correlated with both relational (r = 0.33) and functional (r = 0.26) aspects of the inpatient service. Significant correlations were not found for any of the other demographic variables.

**Table 6 ijerph-19-05574-t006:** Thematic Analysis of Qualitative Consumer Feedback (*n* = 54).

Themes	Qualitative Feedback Quotes	*n*	*%*
**Positive feedback**		40	74.07
Positive experience with staff (overall)	“The staff here are amazing. They are so kind, caring, they really go above and beyond for you and they make sure that all your needs are taken care of. They really are my heroes.”	40	74.07
Support from nursing staff	“The support and friendship of all the nurses has made each day easier than the last, and has made my recovery much more positive.”	21	38.89
Increased self-confidence	“I have gained more confidence with my baby and have been given lots of great tips and support.”	12	22.22
Positive experience with the allied health team	“I was amazed with the care and support I received through allied health, and feel very fortunate to have had such a high level of care.”	8	14.81
Provided good practical skills	“I feel like I got a lot from my stay here. Very practical and centring.”	8	14.81
Positive experience with medical staff	“The doctors were approachable and listened to my care needs.”	7	12.96
Highly skilled and knowledgeable staff	“Thank you for your professionalism and obvious expertise in caring for us women and our babies at a time when we’re not able to take care of ourselves.”	3	5.56
Useful child health nurse visits	“The Child Health Nurse was awesome in telling me techniques to settle the baby and feeding.”	3	5.56
Useful advice offered	“Lavender has taught me not only strategies to cope being a new mum, but really common-sense techniques to help me understand baby better, and to help me bond with baby and genuinely enjoy and embrace motherhood.”	3	5.56
Good ward and facilities	“The facilities were nice and modern.”	3	5.56
Variety of support services available	“With a variety of services ranging from amazing nurses through to physio, I have felt very confident I have come to the right place.”	2	3.70
Social support from other mothers on the ward	“Lovely mothers to learn from and go through the stay together.”	2	3.70
Enjoyable activities on the ward	“Loved the activities and freedom to go for walks.”	2	3.70
Good family involvement	“ [Husband] stayed nearby and spent his days with me at the unit.”	1	1.85
Clear explanations of diagnosis and treatment	“I really feel that time was taken to explain my illness to me. My questions were answered, medications were explained and discussed.”	1	1.85
**Suggestions for improvement**		28	51.85
Better food (including more options for allergies and intolerances)	“More meal options for dairy allergy.”	5	9.26
More MBU facilities	“I really hope [the government] expands to develop more services like Lavender.”	4	7.41
More group sessions and activities (reported as lacking due to COVID-19)	“More ways to connect with other patients, like movie nights.”	4	7.41
Clearer communication between staff and patient related to discharge	“A clearer outline of what happens after discharge before the day of discharge would have been useful.”	3	5.56
Greater family visitations (including partners and older children) during COVID-19	“Letting family come whenever, not restrict hours.”	3	5.56
Increase the temperature in the MBU (facilities reported as too cold)	“Heaters, warmer showers.”	3	5.56
More toys and equipment available for older babies	“Perhaps a playpen to occasionally contain crawling baby.”	2	3.70
Greater availability of allied health professionals	“More child health nurse visits.”	2	3.70
Dissatisfaction with casual nursing staff	“ [Casual staff] don’t appear competent with babies. I did not feel supported [by casual staff].”	2	3.70
More voluntary time outside of the ward	“More voluntary time outside of the ward.”	1	1.85
More cleaning of communal equipment	“The facilities were generally very clean but I would’ve liked it if the cot bars and high chair (especially straps) were cleaned between babies. And maybe if the bath stand was cleaned more regularly, as it did have some built up residue.”	1	1.85
Larger kitchen, bedroom, and bathroom	“Larger kitchen and cooking facilities.”	1	1.85

### 3.5. Thematic Analysis of Qualitative Patient Feedback

Table 6 reports the thematic content analysis of feedback provided by 54 of the participants (77.14%). Forty women (74.07%) detailed positive feedback about the Unit, such as the feedback related to the support from nursing staff (*n* = 21, 38.89%), allied health team (*n* = 8, 14.81%), and medical staff (*n* = 7, 12.96%). All women who provided positive feedback described favorable experiences with the staff (*n* = 40, 74.07%), such as: 


*“The staff here are amazing, so kind and caring, they really go above and beyond to make all your needs are met. They really are my heroes when I was going through something so horrific.”*



*“I am extremely thankful for my stay at Lavender. I was looked after so well and made to feel safe from the very beginning.”*


Twelve women (22.22%) outlined increased self-confidence such as: 


*“Lavender has helped me find my confidence as a mum/wife/woman. On admission I had no hope and felt completely worthless. I am now leaving ready to start a new chapter, full of hope for a brighter future.”*


Six women (11.11%) described that the admission provided good practical skills, such as:


*“I’ve really enjoyed my time at Lavender. I’ve gained more skills with my baby and have been given lots of great tips and support.”*


A range of suggestions for improvement were provided, such as having more MBU facilities (*n* = 4, 7.40%):


*“I really hope [the government] expands to develop more services like Lavender.”*



*“There should be more of these units available.”*


Other suggestions included higher quality food and more choices to cater for allergies and food intolerances (*n* = 5, 9.26%), having more group sessions and activities (*n* = 4, 7.40%), clearer communication between the staff and patient (*n* = 3, 5.56%), greater levels of family involvement (*n* = 3, 5.56%), and increasing the temperature within the MBU (*n* = 3, 5.56%). 

Table 7 displays cross-tabulation to demonstrate the relationship between the type of feedback received and voluntary admission status. Of the women who provided qualitative feedback, 35 were voluntarily admitted to the MBU (64.81%), and ten (18.52%) were involuntarily admitted. Nine of the women (16.67%) provided qualitative feedback anonymously. Of the women who provided positive feedback, 25 (62.5%) were voluntarily admitted, six (15.0%) were involuntarily admitted, and 9 (22.5%) were provided anonymously. Of the suggestions for improvement that were provided, 21 (75.0%) were provided by voluntary patients, five (17.86%) were provided by involuntary patients, and two (7.14%) were provided anonymously. 

**Table 7 ijerph-19-05574-t007:** Cross-Tabulation of Qualitative Feedback Type by Voluntary Admission Status.

	Mental Health Act Status (*n* = 54)		
Feedback Type	Voluntary (*n* = 35, 64.81%)	Involuntary (*n* = 10, 18.52%)	Anonymous (*n* = 9, 16.67%)
Positive Feedback (*n* = 40)	25 (62.5%)	6 (15.0%)	9 (22.5%)
Suggestions for Improvement (*n* = 28)	21 (75.0%)	5 (17.86%)	2 (7.14%)

Note. Frequencies in the table do not equate to the number of participants, given that one participant may have provided both a positive feedback comment and a suggestion for improvement.

## 4. Discussion

This study uses a mixed-method approach to understand the patient experience of being admitted to an MBU, using the POEM and qualitative feedback. Results were largely congruent with hypotheses, which will be discussed below.

This study elucidated that women reported high satisfaction with the experience of functional and relational aspects of the inpatient service, including the usefulness of specific therapeutic groups. Women perceived that staff were sensitive to their needs and demonstrated empathy and active listening. The quantitative results of the present study indicated that women had a strong therapeutic alliance with staff, which is found to be influential in therapeutic outcomes [20]. Qualitative feedback similarly revealed that women appreciated the personal attributes of staff, such as being supportive and caring, and disliked casual nurses (i.e., nurses who are not permanent staff and provide cover when required) for their lack of perinatal-specific knowledge and skills. These findings are congruent with Wright et al. [10] who also found that staff behaviors and attitudes were crucial to the patient experience.

Women also reported that staff provided appropriate psychoeducation about their mental illness to enable them to have an increased understanding of it. Psychoeducation has been found in a comprehensive systematic review to have several benefits such as reducing relapse, increasing medication compliance, improving social function, and lowering anxiety and depression [21].

Women also rated that staff collaborated with them in the treatment process, which is a core tenet of recovery-oriented practice and empowers women to take responsibility for their recovery [22]. This is a strength of the Lavender Unit as women from other perinatal psychiatric inpatient units have requested more involvement in decision-making [12,13]. 

Another aspect appreciated was that staff involved significant others such as family, friends, and relevant community services, which was regarded as being pivotal for the transition from hospital to home. Involving key support people (such as the partner and grandmother of the baby) when in care is a central principle of family-centered care and promotes recovery [23]. Research in other mother and baby units has found that involving partners improved familial relationships and was desired [24,25]. Community services were engaged with consent through liaison during the admission, making referrals for a follow-up post-discharge, collaborating during the discharge planning process, providing a clinical hand over, making face-to-face joint follow-up appointments, and participating in multidisciplinary care reviews. To meet the needs of the mother and baby post-discharge and to prevent relapse, a range of community services is required post-discharge, including engaging private or public psychiatry services, mental health case management, child health QLD and the Early Intervention Parenting Service, public or private infant mental health therapy, perinatal psychology, domestic and family violence non-government organizations, multicultural-specific mental health support services, family support services, Statutory Child Protection services, women’s health services, and targeted support groups, as well as generalist community support for women and children such as play groups, library services, and council exercise programs [16].

In the survey and the open-ended questions, women reported that staff encouraged their relationship with their baby, parenting confidence, self-confidence, and parenting skills, and responded sensitively to their baby’s needs. This is similar to previous findings which suggest that co-admission with the infant improves parenting skills and maternal–infant attachment [10,26].

The timeliness of mental health support was rated favorably in the present study, which is congruent with Antonysamy [14]. The environmental aspects were mostly highlighted as hygienic and clean in the survey feedback; however, aspects to improve upon that were noted in the qualitative feedback included the air-conditioning temperature being too cold; a need for increased cleanliness of communal equipment (e.g., cots and highchairs); more equipment for older babies; and environmental changes such as having larger areas within the unit such as the kitchen, bedroom, and bathroom. Ensuring the MBU space is appropriately designed, comfortable, and appropriate for mother and baby interaction is important, as outlined by Connellan et al. [27].

As overarching indicators concerning the satisfaction with the service, high numbers of women voiced in qualitative and quantitative feedback that they would recommend the service to others, and the majority perceived the MBU as a good place for recovery. Other MBUs have also been well-regarded by the admitted women, and patients have indicated a preference for treatment in MBUs rather than acute psychiatric units without their infant [12,14].

A high proportion of participants (95.71%) rated the Unit as having helpful activities and therapies. Specific therapeutic groups (e.g., pharmacotherapy group, mother and baby exercise) were rated as useful by the majority of women. Of the therapeutic groups outlined, sensory modulation was rated as the most useful by 92.42% of women. In this therapeutic group, women are educated on the zones of arousal and how sensory modulation techniques may be used to support optimal regulation by either calming or alerting their nervous system [28]. Women and babies’ triggers are identified using a parenting-specific checklist, and women are able to trial a range of sensory tools (e.g., fidgets, theraputty, weighted modalities, essential oils) based on their sensory preferences [29]. Women are provided with a ‘sensory kit’ which contains a range of tools and activities that can be used in times of dysregulation when on the Unit and post-discharge [30]. In addition, women complete the Adolescent/Adult Sensory Profile, which assesses mothers’ sensory patterns of sensory sensitivity, sensory avoidance, low registration, and sensory seeking, and categorizes scores based on the normative population (e.g., “much more than most”, “much less than most”) [31]. Sensory modulation techniques are particularly helpful as women with mental illnesses exhibit higher than normal levels of being bothered and overwhelmed by sensory input (i.e., sensory sensitivity and sensory avoidance), which is associated with poorer maternal–infant attachment and lower parenting confidence [29].

There were approximately 7–22% of women who did not participate in particular groups, which may be because of the high acuity of their mental illness and younger aged infants (M = 17.10 weeks or 3.94 months), requiring increased care from the mother (e.g., breastfeeding, settling). Other plausible reasons are that women may be resting are because they are fatigued from waking up throughout the night to care for baby, and/or may be engaging in individual reviews rather than group sessions at that time, that otherwise could not be avoided. Possible contextual reasons for this were the restrictions related to COVID-19, and the unavailability of allied health staff (as indicated in qualitative feedback). 

Results revealed, congruent to hypotheses, that women reported a positive change in mental health status from admission to discharge, regardless of any other clinical or demographic factors. This was consistent with previous research conducted on the Lavender MBU, exploring clinician-rated functional and behavioral improvements based on the Health of the Nation Outcome Scores [16], and the wider literature examining MBUs, indicating improvements in mental health functioning [8,27]. 

Furthermore, the present study identified the demographic variables correlated with satisfaction with the patient experience. Women who were admitted voluntarily were more likely to perceive greater satisfaction with how they were treated by staff compared with women who were admitted involuntarily. Qualitative feedback similarly revealed that an involuntary participant desired more time outside of the ward. Despite this, involuntary patients provided positive qualitative feedback, and patients who were involuntarily and voluntarily admitted reported similar levels of improvement in their mental health from admission to discharge. Voluntary patients are making an informed decision to be admitted, whereas involuntary patients may feel that admission is not warranted as they lack the insight and capacity to consent to medical treatment due to the severity of their mental illness [32]. Similarly, voluntary patients have been found to be more satisfied with inpatient mental health services as patients have had positive therapeutic relationships and have increased insight into their illness [33]. Results of the present study suggest that more targeted interventions towards women who are involuntarily admitted, such as support by the Independent Patient Rights Advisors, may be beneficial. Interestingly, a woman’s age, how long her admission was, her level of social support deemed by her marital status, and level of socioeconomic advantage were not correlated with level of satisfaction, suggesting that equitable care was provided. 

Thematic content analysis of the qualitative feedback revealed a range of improvements within the MBU service, such as a wider range of food options to cater for allergies, intolerances, and dietary preferences (e.g., vegan, halal, gluten-free), which is consistent with qualitative feedback given at other MBU services [14]. Several women advocated for more public MBU beds or services, which is in alignment with recent national position papers suggesting that there is a need for one eight-bedded unit for every 15,000 deliveries, which is not currently met [34,35]. Women also voiced the need for more therapeutic group activities and visitations from family members that had been restricted during the COVID-19 pandemic. Soh et al. [36] similarly identified that COVID-19 affected clinical care and relationships. In the present study, patients desired more discharge planning and preparation, which has been found to promote the transition to home in qualitative studies [37]. Overall, there were more positive comments than suggestions for improvement to the service, which was consistent regardless of whether the women were voluntarily or involuntarily admitted. 

### Limitations

This study used a single-site design, with future studies recommended to explore multiple MBUs. Another limitation was that the first question of the POEM, examining mental state at admission, was open to recall bias. Future research may investigate updating the POEM questionnaire as not all qualitative feedback raised (e.g., size of the Unit, social support from other patients, COVID impact) were captured in the survey. This, however, also highlights the benefits of supplementary qualitative feedback. Further areas of research that may be explored include examining satisfaction feedback post-discharge or at multiple timepoints, perspectives from a key support person (e.g., partner, grandmother of baby), and triangulation with psychometric measures such as the Edinburgh Postnatal Depression Scale.

## 5. Conclusions

This study used both quantitative and qualitative methodologies to understand the patient experience and outcomes of admission to an inpatient MBU. Findings suggested that women were largely satisfied with the care provided, such as the responsiveness of staff towards themselves and their baby, and the support provided in relation to maternal-infant attachment and parenting skills. The study also highlighted areas of improvement within the MBU service, such as better food quality, more availability of public MBU beds, training for casual nurses, environmental design, infection control management during COVID-19, and supporting patients who were involuntarily admitted. As this feedback from patients informs service development, the study underpins the importance of partnering with patients to deliver the highest quality, patient-centered care.

## Data Availability

Data is available upon request from the author.

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
