# Peer review of "Evaluation of an Inpatient Psychiatric Mother-Baby Unit Using a Patient Reported Experience and Outcome Measure"

_ijerph, 2022, doi:10.3390/ijerph19095574_

Round 1
Reviewer 1 Report
It is very concerning that this study did not undergo ethical review, particularly as it involves vulnerable women. There was obvious linkage between the survey results and patient records and with the inclusion of Aboriginal women makes it crucial that ethics are attained. If exemption is appropriate this needs to be fully explained by the authors.
The study refers to 70 women but the breakdown shows 69 women - which is it.
There is inconsistent use of zipcode and postcode.
There was no investigation of the proportion of the 40 qualitative comments received as correlated to voluntary admissions - this is an important issue to untease and compare.
These issues need to be addressed before publication is considered.
Author Response
Thank you for your comments. Please see the attachment for our response.

Reviewer 2 Report
This study aims to explore the patient experience of an mother and baby unit (MBU) admission. Specifically, they aimed understanding a series of qualitative and quantitative aspects related to the admission into a MBU. These were, the experience of admission to an MBU, including the usefulness of therapeutic groups; to test for the change in perceived mental health from admission to discharge; to evaluate demographic variables associated with patient experience; fourthly, to examine qualitative feedback given by patients prior to discharge. The study is of interest and I have some suggestions for the authors.
Specifically:
- The authors state that “Between January 2019 and January 2021 (24 months), a sample of 70 women were 93 recruited for the study from the MBU.” Were these consecutive patients?
- The authors state that “All women who were admitted to the MBU were eligible to participate.” The inclusion and exclusion criteria are not reported.
- The authors found a statistically significant increase in patients’ perception of mental health status from admission to discharge. What about variables associated with this improvement?
- It is not clear whether some mothers had mental health problems. Description of diagnoses?
- In case of mental health problems is the service linked with community mental health services or equivalent?
Author Response
Thank you for your comments. Please see the attachment for our response to your feedback.

Reviewer 3 Report
Dear authors,
this research you have conducted i of great importance for improving quality of life of mothers who suffering from postpartum depression and other mental/psychological conditions and disorders.
This paper is very well organized, and I don't have any objections on methodology of research.
only if the paper can be published as a professional paper. But in scientific terms the work has a number of shortcomings and I would choose to reject.As a professional paper it has all the relevant features necessary for publication. But in order to be considered as a scientific work, the methodology itself does not have all necessary elements. In the case of such a convenient sample, observation at one point in time does not contribute to a better understanding of the problem; It provides only answers and explanations for this observed group of respondents; the work is more on a descriptive level.
Sufficient information are presented in introduction part and explanations of your research data are clearly presented.
I have nothing to add.
Author Response

(The authors gave the same response as above.)
